# Assessment of heavy metal pollution in sediments from the urban section of Yihe River, Linyi City, China

Qinghai Deng[1], Guizong Sun[1], Fuquan Li[2], Lei Guo[2], Dan Shi[2], Liping Zhang[1]*, Zhenzhou Sun[1], Jingjing Yang[1]

1 Collage of Earth Sciences and Engineering, Shandong University of Science and Technology, Qingdao, Shandong Province, China, 2 The Seventh Geological Brigade of Shandong Provincial Bureau of Geology and Mineral Resources, Linyi, Shandong Province, China

* zhangliping@sdust.edu.cn

## Abstract

The Yihe River is the largest mountain torrent river in Shandong Province, which plays an important role in flood control, irrigation and climate regulation. Due to the impacts of the upstream and urban domestic sewage, as well as industrial and agricultural wastewater, the Linyi City section of the Yihe River is expected to have a high risk of metal pollution. Sediments are the main reservoir and potential release source of metals (metalloid) in river systems. Assessment of metals in sediments can identify anthropogenic pollution. In this study, 25 sediment samples were collected from the Linyi City section of the Yihe River and its tributaries, and the concentration of As, Cd, Cr, Cu, Ni, Pb and Zn were quantified by inductively coupled plasma mass spectrometry (ICP-MS), and the concentration of Hg was determined by atomic fluorescence spectrometry (AFS). The pollution levels were evaluated by determining the contamination factor, pollution load index, geoaccumulation index, potential ecological risk assessment and toxicity risk index. Correlation analysis and absolute principal component-multiple linear regression (APCS-MLR) were used to conduct source apportionment. Cr, Cu, Pb, Zn, Ni, Cd, As, and Hg were detected in all sediment samples. Overall, the concentration of metals (metalloid) in the sediments of the main stream of the Yihe River is mostly within the environmental background value, and the overall state is from no pollution to slightly polluted, while the tributaries of the Yihe River are in a slightly polluted state. Hg and Cd are the two main metal pollutants in the surface sediments of the study area, with the average content of 1.65 and 1.11 times the background value, respectively. Most areas of the main stream of the Yihe River are free of metal pollution, with low ecological risk and no toxicity risk. The ecological risks in the tributaries (Suhe River, Benghe River, Liuqinghe River) and the river center island (Yihe River North Island) are high and assessed as presenting low toxicity. Source analysis showed that Cr, Ni, Cu, Zn, and As mainly come from natural sources and agricultural activities, while Cd, Pb, and Hg are mainly the result of transportation and industry. The results help us understand the distribution and pollution of metals (metalloid) in the river sediments, and also provide management support to local environmental management departments and relevant national departments.

**Data availability statement:** All relevant data are within the manuscript and its Supporting information files.

**Funding:** This research was financially supported by the Shandong Provincial Natural Science Foundation (Grant No. ZR2022MD032). The funding agency had no role in study design, data collection and analysis, decision to publish, or preparation of the manuscript. The authors are grateful for the support.

**Competing interests:** The authors have declared that no competing interests exist.

## Introduction

Metals (metalloids) come from a wide range of sources, are highly toxic, persistent and non-biodegradable, thus have attracted increasing attention from researchers [1–3]. In recent decades, the exponential growth of the global population and economy, as well as the development of industry, agriculture and urbanization, have jointly led to the introduction of large amounts of metals into rivers, resulting in the decline of water quality, and increased metal pollution and ecological risks [4–6]. Metals can be enriched in sediments by physical adsorption, chemical precipitation and aquatic degradation. Sediments can store and accumulate metals from industrial emissions, agricultural activities, urban sewage discharge, and atmospheric deposition for a long time, making the metal content in water bodies lower. When disturbed or hydrodynamic conditions change, metals in sediments will re-enter the water body as the sediment-water interface environment alters and become a source of secondary pollution [7–9]. Metals can also have a negative impact on aquatic ecosystems through bioaccumulation and biomagnification processes. If local residents continue to be exposed to metals exceeding the safe limit, it may have harmful effects and lead to non-carcinogenic risks such as neurological diseases, liver and kidney diseases [9–12]. Children in particular, whose physical development is not yet fully mature, have relatively weak functions of their organs and systems, and have limited ability to metabolize and excrete metals. Sediments in river systems are more representative of their pollution status [13] and have become valuable environmental archives for identifying anthropogenic pollution and metal enrichment in river environments [14]. The assessment of sediment quality can provide information on pollution and environmental conditions in aquatic ecosystems [15]. Therefore, studies on the characteristics, source factors and ecological effects of metal pollution in sediments are crucial to better control metal pollution in the environment [16].

Yihe River, a major tributary of the Yishu-Si River system within the Huaihe River Basin, stretches across the southern part of Shandong Province and the northern part of Jiangsu Province in southeast China, with a total length of 574 kilometers. Within this system, the Linyi section spans 226 kilometers and flows through Yishui County, Yinan County, urban areas, Lanling County, and Tancheng County in Linyi City. As the largest river in Linyi City, the Yihe River has a complete ecosystem structure and function, rich biodiversity, and along with its tributaries, it plays an important role in flood control, irrigation, climate regulation, and maintaining ecological balance. However, the urban section of the Yihe River and its tributaries in Linyi City are seriously affected by domestic sewage, industrial wastewater and agricultural non-point source pollution from upstream counties and three districts under the jurisdiction of Linyi City [17], which may become potential sources of metals in the sediments of the Yihe River. Previous studies mainly analyzed the characteristics of metals pollution in the sediments of the Yihe River [18–20], but did not address the source of metals pollutants in the sediments of the urban section of the Yihe River and its tributaries in Linyi City. Therefore, the objectives of this study were: (1) to describe the distribution characteristics of eight metals (Cd, As, Pb, Zn, Cu, Hg, Ni, Cr, Cd) in sediments; (2) to evaluate the pollution status, ecological risks; and (3) to determine the possible sources of metals through APCS-MLR and Pearson correlation analysis. The results are expected to enhance our understanding of metal pollution in the Yihe River and provide a reference for pollution control and ecological restoration by relevant local departments.

## Materials and methods

### Overview of the study area

The Yihe River, located in eastern China, originates from the northern foot of Niujiao Mountain in Yiyuan County, Zibo City, Shandong Province. It flows through Yishui

County, Yinan County, Mengyin County, Pingyi County, as well as other counties in Linyi City, and into the Yellow Sea from Jiangsu Province. It has a total length of 574 km and a drainage area of 17325 km². The Yihe River Basin has a temperate continental monsoon climate with four distinct seasons, and rain and heat occurring at the same time. The annual average temperature is 13.4 ~ 15°C, and the average annual rainfall is 800 ~ 848 mm. Three tributaries, including the Benghe River, Liuqinghe River, and Suhe River, merge into the Yihe River in the northeast of Linyi City [21]. In the 1990s, Linyi City built the longest Xiaobudong Rubber Dam in Asia on the main stream of the Yihe River for water storage, forming a vast water surface of 1066.67 hm² (also known as Yimeng Lake) in Linyi City [20]. Due to the rapid development of industry and agriculture in Linyi in recent decades and the accelerated increase in the population, the Yihe River and its tributaries may be contaminated by metals [17,18,21].

## Sample collection and analysis

In February 2023, a total of 25 sediment samples were collected in the Yihe River Basin from the 0–20 cm layer (Fig 1). Sampling was performed using a portable sediment sampler. Three samples were collected from the area around each sampling point and mixed evenly to ensure statistical stability. The mixed samples were placed in clean polyethylene bags, sealed and numbered, and stored in a refrigerator at 4°C. The geographic location was recorded using a GPS device. The sediment was air-dried at room temperature, made into powder, and sieved through a 100-mesh screen to remove impurities such as plant residues and gravel in the sediment. After grinding, it was stored in a polyethylene bag for later use. No special permission was required to take samples from the Yihe River. The study area was not privately owned or protected in any way, and the field research did not involve endangered or protected species.

All analyses were performed at the laboratory of The Seventh Geological Brigade of Shandong Provincial Bureau of Geology and Mineral Resources. According to the method of "Solid Waste - Metal Inductively Coupled Plasma Mass Spectrometry Determination" (HJ766-2015), the concentrations of As, Cd, Cr, Cu, Ni, Pb and Zn in the sediments of the Yihe River were quantified by inductively coupled plasma mass spectrometry (ICP-MS). Briefly, 0.10 g of sediment powder was taken, HCl (1.0 mL), HNO₃ (4.0 mL), HF (1.0 mL),

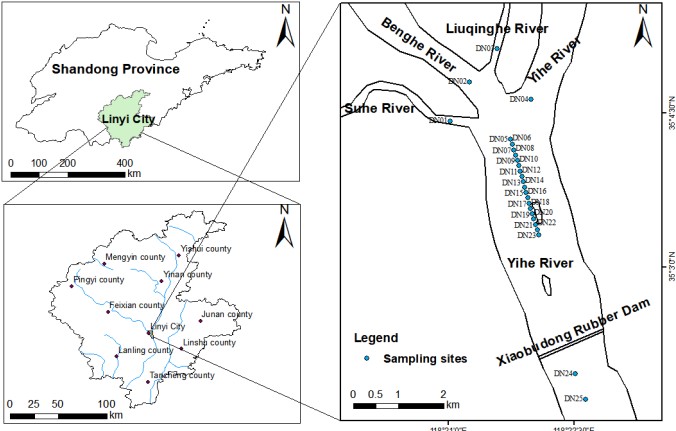

**Fig 1. Geographical location and sampling points of the study area.** (Map data comes from Natural Earth. http://www.naturalearthdata.com/).

and $H_2O_2$ (1.0 mL) were added, the internal solutes were dissolved and transferred to a volumetric bottle, and deionized water was added for the volume to reach 50 mL as the sample. The concentration of metals was measured using an inductively coupled plasma mass spectrometer (iCAP RQ, USA). According to the "Soil and Sediment - Determination of Mercury, Arsenic, Selenium, Bismuth, Antimony - Microwave Dissolution/Atomic Fluorescence Method" (HJ680-2013), the concentration of Hg in soil samples was determined by atomic fluorescence spectrometry (AFS). To 0.50 g of sediment sample, aqua regia (5.0 mL) and distilled water (5.0 mL) were added, and after digestion, it was transferred to a volumetric flask and diluted to 50 mL by adding deionized water as the sample. The Hg concentration was measured using an atomic fluorescence spectrometer (AFS-8230, China).

To ensure high accuracy, strict quality management and control were adopted for all experimental procedures. A certified reference sediment (GBW07401) was used to determine the analytical accuracy, with relative standard deviations (RSD) of < 5% for all samples. The recovery rates for all metals ranged from 95% to 105%. In addition, the limits of detection (LOD) of each metal were different (Table 1).

However, it should be noted that due to research design and resource limitations, no water samples were collected for testing in this study. Therefore, we were unable to directly analyze the relationship and differences between metals in water and sediments. Nevertheless, through detailed sediment analysis, we still revealed some important characteristics of metal pollution in the area.

## Pollution assessment methods

**Contamination factor (*CF*) and pollution load index (*PLI*).** The pollution situation of a harmful substance can be evaluated by calculating the contamination factor (*CF*) [22]. The calculation formula is as follows (Equation 1):

$$CF = C_{metal} / C_{background} \tag{1}$$

where $C_{metal}$ refers to the actual measured metal concentration in the sediment at a specific location; $C_{background}$ is the metal background value. Based on the *CF*, metal pollution can be divided into four groups: *CF* < 1: low pollution; 1 < *CF* < 3: moderate pollution; 3 < *CF* < 6: considerable pollution; *CF* > 6: high pollution [23].

The Pollution Load Index (*PLI*) is a comprehensive measure that represents information on different elements within an area. *PLI* can reflect spatiotemporal changes and the contribution of each element to total pollution [24]. It is calculated as follows (Equation 2):

$$PLI = (CF_1 \times CF_2 \times CF_3 \times ..... \times CF_n)^{1/n} \tag{2}$$

where n denotes the total number of metal elements detected in the sediment sample. Based on *PLI*, samples can be divided into four categoris: *PLI* < 1, unpolluted; 1 ≤ *PLI* < 2, moderately polluted; 2 ≤ *PLI* < 3, highly polluted; *PLI* > 3, extremely polluted [25].

Table 1. Limits of detection of metals (mg kg⁻¹).

|  | Cr | As | Cu | Hg | Ni | Pb | Zn | Cd |
|---|---|---|---|---|---|---|---|---|
| LOD | 1.5 | 0.2 | 0.1 | 0.0005 | 0.2 | 0.2 | 1 | 0.02 |

**Geoaccumulation index ($I_{geo}$).** The $I_{geo}$ index was proposed by Müller [26]. It comprehensively considers the impact of human activities and natural geological processes on the background value of metals, and is calculated using the Equation 3:

$$I_{geo} = \log_2 \frac{C_n}{1.5 \times B_n} \tag{3}$$

where $C_n$ represents the measured concentration of metals in sediments; $B_n$ is the geochemical background concentration [27]. Due to the change in lithology, the factor 1.5 is added to the formula as the background matrix correction factor [28]. Based on the $I_{geo}$ values, samples can be divided into different categories: $I_{geo} < 0$: uncontaminated; $0 < I_{geo} < 1$: uncontaminated to moderately polluted; $1 < I_{geo} < 2$: moderately contaminated; $2 < I_{geo} < 3$: moderately to strongly contaminated; $3 < I_{geo} < 4$: strongly contaminated; $4 < I_{geo} < 5$: strongly to extremely contaminated; $I_{geo} > 5$: extremely contaminated [29,30].

## Risk assessment methods

**Potential ecological risk assessment.** The potential ecological risk index can better evaluate the potential ecological risk factors of metal pollution by combining ecological and environmental effects with toxicology [31]. The following equation is used (Equations 4, 5 and 6) [22]:

$$E_r^i = T_r^i C_f^i \tag{4}$$

$$C_f^i = C_n^i / C_o^i \tag{5}$$

$$RI = \sum E_r^i \tag{6}$$

where $E_r^i$ represents the potential ecological risk of a single element; $RI$ represents the sum of the risk aspects of all metals in the sediment; $T_r^i$ represents the toxicity response factor; $C_n^i$ represents the metal concentration; $C_o^i$ is the reference value in this study. The coefficients of elemental toxicity $T_r^i$ are: Hg = 40, Cd = 30, As = 10, Cu = Pb = Ni = 5, Cr = 2, Zn = 1. The calculated $E_r^i$ and $RI$ value classification is shown in Table 2 [22,32].

**Toxicity risk index ($TRI$).** The $TRI$ is based on the $TEL$ and $PEL$ effects to evaluate the toxicity risk of metals in sediments [28]. The following formula is used for the Equation 7:

$$TRI_i = \sqrt{\frac{(C_i / TEL)^2 + (C_i / PEL)^2}{2}} \tag{7}$$

**Table 2. Classification of potential risk index.**

| $E_r^i$ | Single factor risk level | $RI$ | Potential risk level |
|---|---|---|---|
| $E_r^i < 30$ | Low risk | $RI < 100$ | Low |
| $30 \leq E_r^i < 50$ | Medium risk | $100 \leq RI < 150$ | Medium |
| $50 \leq E_r^i < 100$ | Strong risk | $150 \leq RI < 200$ | Strong |
| $100 \leq E_r^i < 150$ | Very high risk | $200 \leq RI < 350$ | Very high |

The comprehensive toxicity risk is calculated using the following Equation 8:

$$TRI = \sum_{i=1}^{n} TRI_i \tag{8}$$

where $TRI_i$ represents the toxicity risk index of a single metal; $C_i$ represents the concentration of metals in sediment samples; $n$ represents the number of metals; $TRI$ represents the toxicity risk index; $TEL$ represents the threshold effect level; $PEL$ represents the possible impact level. The $TRI$ values are divided into different categories: $TRI \leq 5$, no toxicity risk; $5 < TRI \leq 10$, low toxicity risk; $10 < TRI \leq 15$, moderate toxicity risk; $15 < TRI \leq 20$, considerable toxicity risk; $TRI > 20$, very high toxicity risk.

## Statistical analysis

Correlation analysis and principal component analysis are both basic mathematical analysis methods. The former refers to the analysis of two or more correlated variables to measure the degree of correlation between the two variables. It can illustrate the degree of correlation between the eight metals and the relationship between their sources. A stronger correlation indicates that the sources are closer. The hierarchical analytic approach is a systematic method that decomposes complex multi-objective decision-making problems into multiple levels and conducts decision analysis by combining qualitative and quantitative methods. The normality test was performed using the Shapiro-Wilk test, and the significance results were all less than 0.05, indicating that the results were significant and the null hypothesis was rejected. The data did not follow a normal distribution. Therefore, Pearson correlation analysis cannot be used, while Spearman correlation analysis was suitable for use. In this study, Spearman correlation coefficient was used to explore the correlation between metals. To explain the correlation between metal concentrations in the Yihe River, IBM SPSS 27.0 (SPSS Inc., Chicago, IL, USA) was used to calculate the Spearman correlation matrix, hierarchical analytic approach and principal component analysis.

Spearman correlation analysis is a non-parametric statistical method used to evaluate the monotonic relationship between two variables. The specific steps are as follows. First, sort the data of each variable according to the numerical size. If the numerical values are the same, they are given the same rank. Next, assign a rank to each data point. If there are missing values or outliers in the data set, they can be excluded from the rank assignment. Calculate the difference between each pair of variable rankings. Use the Spearman correlation coefficient formula to calculate the correlation coefficient.

The APCS-MLR receptor model was employed to quantify the average contribution rate of pollution sources to metals, thereby quantitatively describing the contribution of pollution sources to elements. Plotting was performed utilizing Origin 2022 (OriginLab Corp., Massachusetts, USA). The spatial distribution map of different metals was created using the inverse distance weighted (IDW) package in ArcGIS 10.8. IDW is a geospatial interpolation technique that predicts the values of variable parameters at locations around a sample site.

## Results

### Distribution of metals in surface sediments

The analytical results for the 25 sediment samples in the study area are shown in Table 3, and the distribution of specific metals is shown in S1 Table. The metal elements Cr, Cu, Pb, Zn, Ni, Cd, As, and Hg were detected in all sediment samples. The pH of the sediments in the study area ranged from 7.83 to 8.42, with an average of 8.26, indicating that the sediments of

**Table 3. Statistical characteristics and relevant standards of metal mass concentrations in sediments of the Yihe River Basin (mg kg⁻¹).**

|  | Cr | Ni | Cu | Zn | Cd | Pb | As | Hg | pH |
|---|---|---|---|---|---|---|---|---|---|
| Average value | 36.72 | 16.22 | 9.89 | 58.73 | 0.12 | 20.79 | 2.26 | 0.04 | 8.26 |
| Maximum | 74.97 | 40.18 | 37.50 | 173.31 | 0.61 | 45.28 | 7.93 | 0.44 | 8.42 |
| Minimum value | 12.83 | 4.51 | 2.29 | 12.32 | 0.025 | 15.18 | 0.80 | 0.005 | 7.83 |
| Median | 34.05 | 13.29 | 5.73 | 39.66 | 0.08 | 18.43 | 1.41 | 0.01 | 8.29 |
| Standard deviation | 15.00 | 11.21 | 9.89 | 51.08 | 0.14 | 6.63 | 2.08 | 0.09 | 0.13 |
| Coefficient of variation | 0.41 | 0.69 | 1.00 | 0.87 | 1.13 | 0.32 | 0.92 | 2.20 | 0.02 |
| Environmental background value | 56.2 | 23.5 | 19.6 | 56.1 | 0.11 | 25.41 | 7.05 | 0.024 | – |
| ERL value of American sediment quality standard | 81 | 20.9 | 34 | 150 | 1.2 | 46.7 | 8.2 | 0.15 | – |
| TEL Value of Canadian Sediment Quality Standard | 52.3 | 18 | 18.7 | 124 | 0.7 | 30.2 | 7.24 | 0.13 | – |
| Average shale value [27] | 90 | 68 | 45 | 95 | 0.3 | 20 | 13 | 0.4 | – |

the Yihe River are alkaline. The concentrations of Cr, Ni, Cu, Zn, Cd, Pb, As, and Hg were 12.83–74.97, 4.51–40.18, 2.29–37.50, 12.32–173.31, 0.025–0.61, 15.18–45.28, 0.80–7.93, and 0.005–0.44 mg kg⁻¹, respectively. The average concentration order was Zn (58.73 mg kg⁻¹)> Cr (36.72 mg kg⁻¹)> Pb (20.79 mg kg⁻¹)> Ni (16.22 mg kg⁻¹)> Cu (9.89 mg kg⁻¹)> As (2.26 mg kg⁻¹)> Cd (0.12 mg kg⁻¹)> Hg (0.04 mg kg⁻¹). The maximum values of Hg and Ni far exceeded the standards of ERL and TEL, but the average values were within the standard limits, indicating the presence of point source pollution. Taking the environmental background values in Shandong Province as a reference, the concentrations of metals in some sampling points of sediment in the Yihe River Basin exceeded the background values, with the maximum values of Cr, Cu, Pb, Zn, Ni, As, and Hg being 1.33, 1.71, 1.91, 3.09, 1.78, 1.12, and 18.33 times the environmental background values, respectively. The maximum values of Zn and Hg exceeded the standard significantly, with the excess rates of Cr, Cu, Pb, Zn, Ni, Cd, As, and Hg at 16%, 20%, 16%, 28%, 32%, 20%, 8%, and 28% respectively. The average concentrations of Hg, Zn, and Cd were 1.65, 1.05, and 1.11 times the background value, indicating a certain metal pollution in sediments in the study area. The values of Cr, Ni, Cu and As were all below the average shale values, but the Zn, Cd, Pb and Hg at some points exceeded the average shale values, which also indicated the presence of pollution.

The IDW interpolation method was used to study the distribution characteristics of Cr, Ni, Cu, Zn, Cd, Pb, As, and Hg in the Yihe River (Fig 2). The results show that the distribution of metals in the study area exhibits certain spatial differences, and the distribution patterns of the eight metals are similar. Overall, the concentrations of As, Cu, Ni, Pb, Ni, Pb, and Zn are low in most areas of the main stream of the Yihe River, while there are two places where the concentrations are significantly higher than other points, namely, sampling points DN05 and DN23. Compared with the main stream of the Yihe River, the eight metals show higher concentrations in the Yihe River tributaries Suhe (DN01), Benghe (DN02), and Liuqinghe River (DN03).

## Evaluation of metal pollution

The contamination factor (*CF*) is a clear representation of the distribution of trace elements. The average values of the single factor pollution index of the eight metals measured in the sediments of the Yihe River, ranked from large to small, were Hg (1.65)> Cd (1.11)> Zn (1.05)> Pb (0.82)> Ni (0.69)> Cr (0.65)> Cu (0.50)> As (0.32). According to the classification standard, Hg, Cd, and Zn are in a medium pollution state, and Pb, Ni, Cr, Cu, and As are in a non-polluted state, indicating that Hg, Cd and Zn are more likely to have adverse ecological

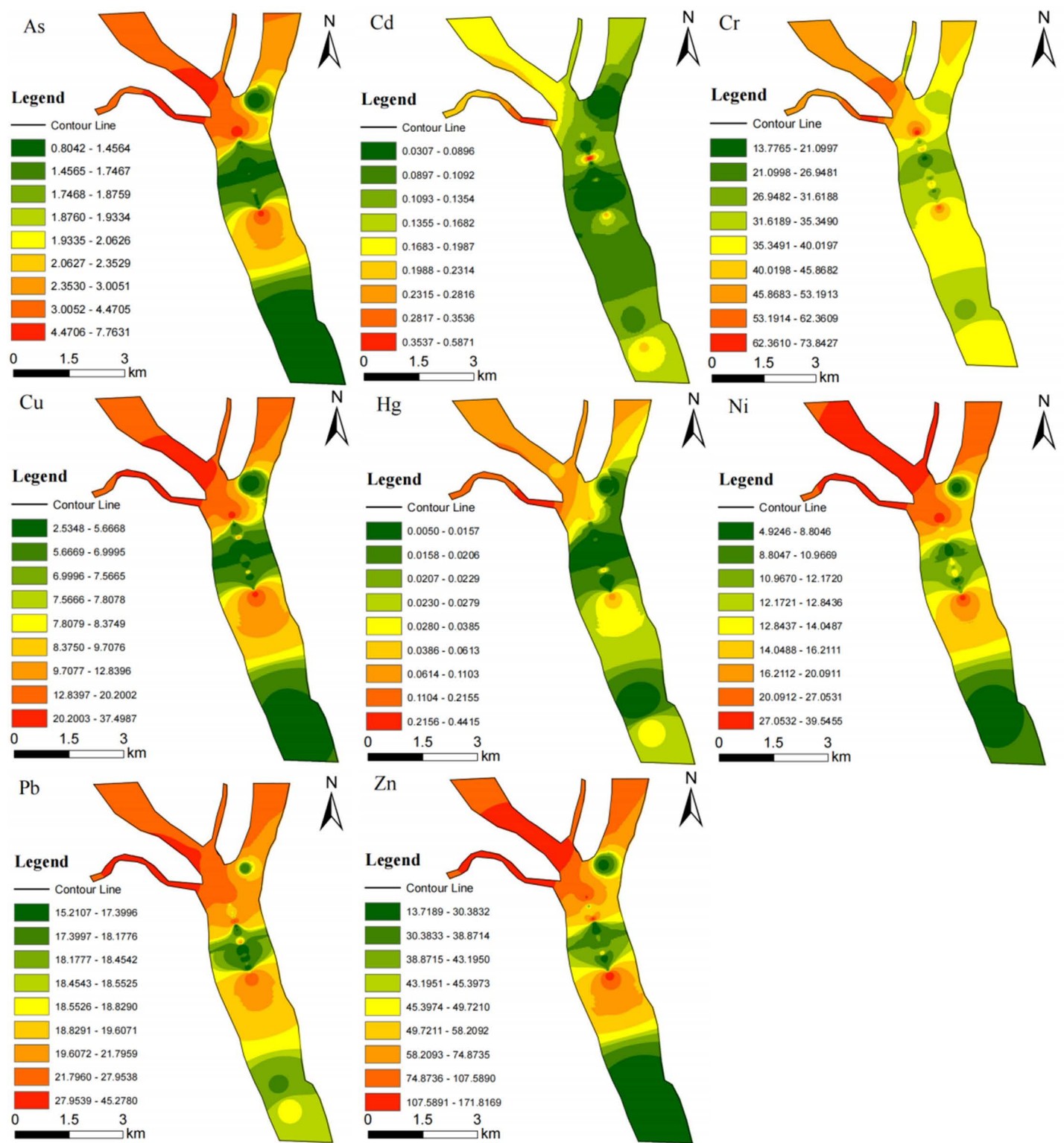

**Fig 2. Spatial distribution of metal parameters.**

effects. The *PLI* of sediments is generally used to estimate the pollution level associated with multiple pollutants in a specific location [33]. According to the displayed *PLI* values (Fig 3a), the pollution levels of different sampling points vary greatly. DN01 is highly polluted (*PLI* = 2.49), DN02, DN03, DN05, and DN23 are in a moderately polluted state. The probability index distribution of *PLI* shows that 96% of *PLI* is below 1.65, which means it is no pollution to moderate pollution.

Fig 4a presents the box plot of the average $I_{geo}$ values for 8 metals. The $I_{geo}$ values for Cr, Ni, Cu, Zn, Cd, Pb, As, and Hg ranged from -2.72 to 0.17, -2.97 to 0.19, -3.69 to 0.35, -2.77 to 1.04, -2.72 to 1.89, -1.33 to 0.25, -3.72 to 0.42, -2.84 to 3.62, respectively. The values are arranged in descending order of the average values as follows: Pb (-0.93)> Zn (-0.98)> Cd (-1.02)> Hg (-1.06)> Cr (-1.36)> Ni (-1.41)> Cu (-2.06)> As (-2.60). The difference in $I_{geo}$ shows that the concentration of metals in the Linyi City section of the Yihe River varies greatly, and there is an obvious point source pollution. The different colors in (Fig 4b) represent the $I_{geo}$ category of each metal at DN01 ~ DN25. It can be seen that Hg pollution has great spatial variation. The degree of Hg pollution is the highest at DN01, where $I_{geo}$ reaches 3.62, which is serious pollution, while it shows moderate pollution at DN23. The $I_{geo}$ of Hg at other sampling points is below 1, indicating no pollution to mild pollution. The pollution of Zn and Cd reaches the maximum at DN01, reaching moderate pollution, and it shows no-pollution ~ mild pollution at other sampling points. The $I_{geo}$ of Ni, Cu and Pb is between 0 and 1, indicating no pollution to mild pollution; Cr and As show no pollution at all sampling points, and $I_{geo}$ is negative. Overall, the $I_{geo}$ values of Cu, Zn, Cd, and Hg at DN01 are higher than those at other sampling points, indicating that metal pollution at DN01 is more serious than that at other sampling points.

## Metal risk assessment

The comprehensive potential ecological risk index (*RI*) and toxicity risk index (*TRI*) of Yihe River were calculated and shown in Fig 5. It can be seen that there are large differences in the

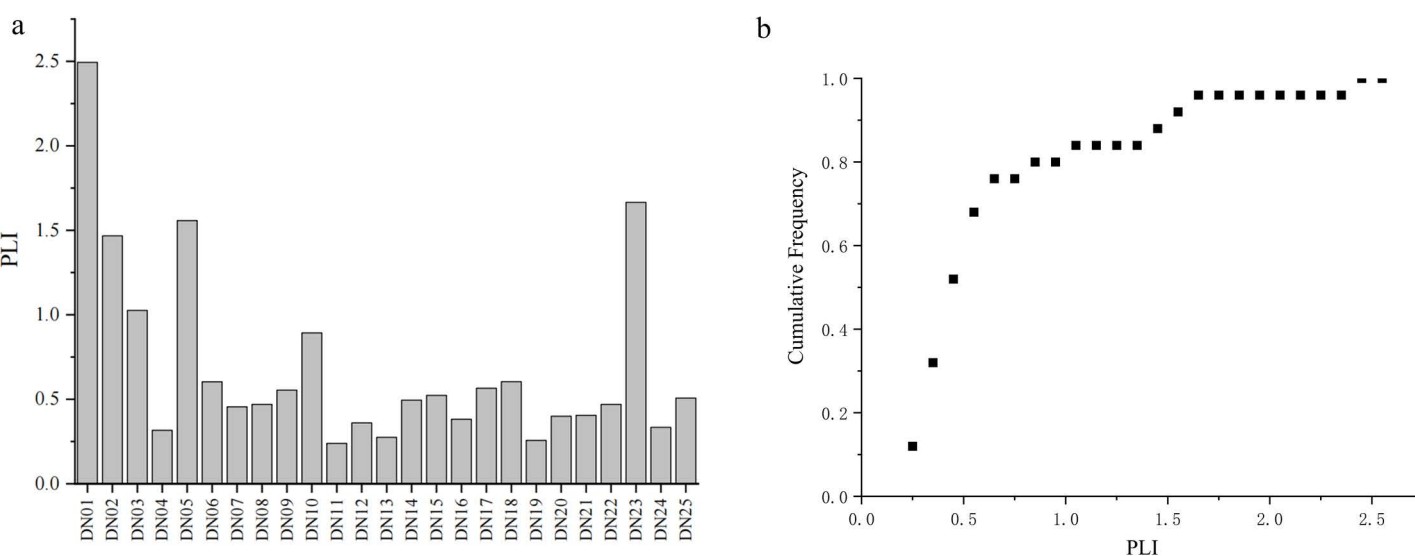

**Fig 3. *PLI* distribution of metal (a) and probability index distribution of *PLI* (b).**

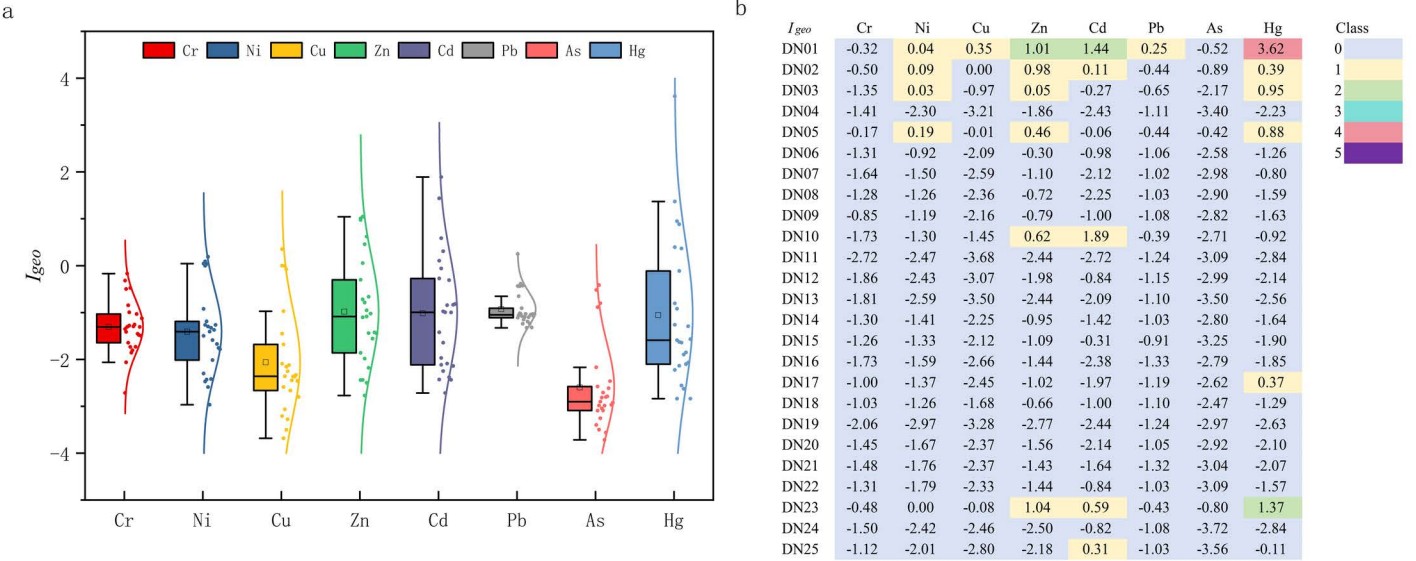

**Fig 4.** (a) $I_{geo}$ box plot of metals in the sediments and (b) block diagrams of pollution categories at different sampling points.

*RI* and *TRI* values at various points in the Yihe River, which is also the case for single-factor potential risks. Specifically, the potential ecological risk indices of Cd and Hg vary greatly, ranging from no ecological risk to extremely high ecological risk, with average values of 33.45 and 65.90, respectively. The *Er* values of the other six metals (Cr, Ni, Cu, Zn, Pb, As) are relatively low, with average values of 1.31, 3.45, 2.52, 1.05, 4.09, and 3.20, respectively, indicating low ecological risks. As can be seen from Table 1, Hg and Cd account for very little content in the sediments. However, due to their high toxicity, these two metals occupy a much larger proportion in *RI* than other metals and are the main potential pollution risk factors. Cd pollution is the most serious and brings extremely high ecological risks at DN01 and DN10, strong ecological risks at DN23 and DN25, and medium ecological risks at DN02, DN03, DN05, and DN15. The ecological risk of Hg at DN01 is as high as 735.94, and it is also extremely high at DN03, DN05, and DN23. Overall, the sampling points DN01, DN10 and DN23 are characterized by extremely high potential ecological risks, DN02, DN03 and DN05 carry strong ecological risks, DN17 and DN25 have medium ecological risks, and other sampling points have low risk levels. In Fig 5b, the distribution of toxicity risk coefficient *TRI* is slightly different. The overall assessment is that DN01, DN02, DN05, and DN23 have low toxicity, and other sampling points have no toxicity. Both risk indices show high risks at the tributary sampling points (DN01, DN02), tributary intersections (DN05), and river islands (DN23).

## Discussion

The coefficient of variation (CV) is the ratio of standard deviation to the mean, which can eliminate the influence of unit difference and mean difference [6]. It can be used to analyze the impact of anthropogenic activities on environmental dynamics [8]. According to the CV, the level of variation can be divided into low variation (CV ≤ 0.20), medium variation (0.20 < CV ≤ 0.50), high variation (0.50 < CV ≤ 1.00), and extremely high variation (CV ≥ 1.00) [34]. Based on our calculations, the CVs of Ni, Cu, Zn, Cd, As, and Hg in the study area are all high and extremely high, and the CVs of Cr and Pb are less than 0.5, which are medium

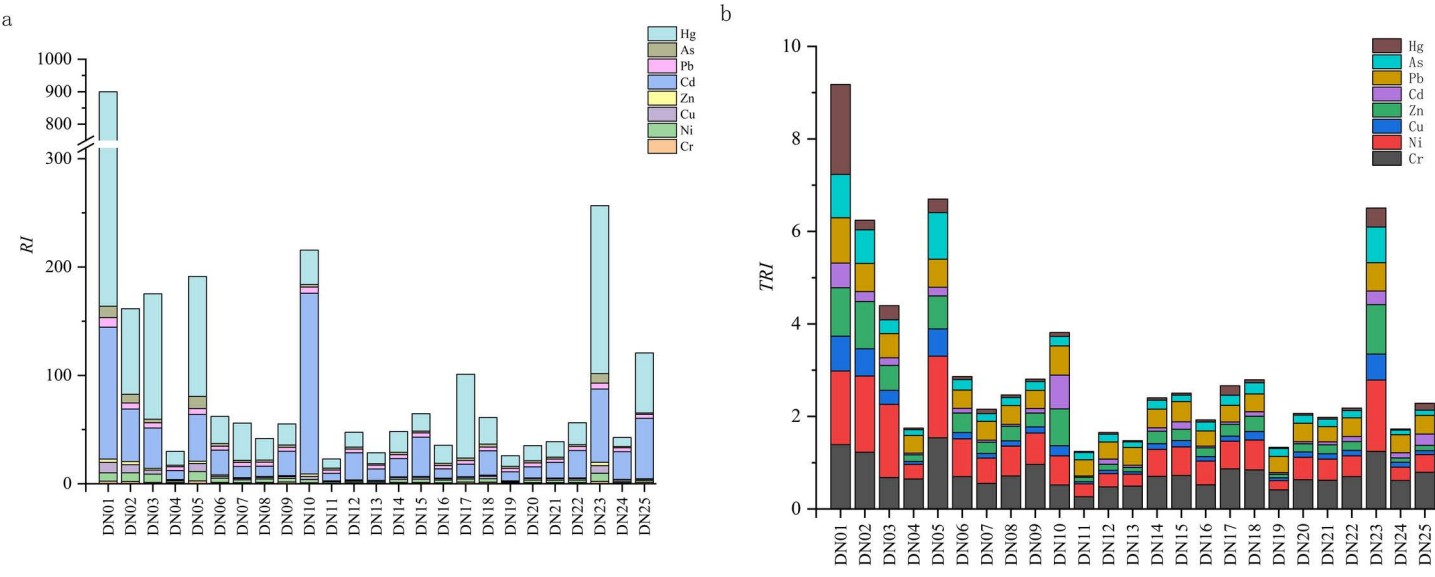

**Fig 5.** *RI* **values of metal elements (a) and** *TRI* **distribution (b).**

variations, indicating that the content of metals in the sediments of the Yihe River is greatly affected by anthropogenic activities.

The global Moran's index is used to describe the overall spatial autocorrelation of metal distribution in the entire study area. Its value range is -1 to 1, and global Moran's I > 0 indicates an clustering trend, and global Moran's I < 0 indicates a dispersed direction [35]. The spatial distribution characteristics of *RI* and *TRI* in the study area were evaluated using the global Moran's I, and the results are displayed in Fig 6. The results show that the values of global Moran's I are 0.106 and 0.332, respectively. The values are > 0 (P < 0.05), and the spatial distribution shows a clustering trend, indicating that *RI* and *TRI* may be affected by external factors. And high-high cluster in tributaries indicates that the metal pollution level is higher there.

On a spatial scale, the distribution of metals in Yihe River sediments varies greatly (Fig 2). metals are concentrated in tributaries and near the central island of the Yihe River, indicating that human activities in this area strongly impact metal accumulation. This is consistent with the *CF*, *PLI*, $I_{geo}$, *TR*, and *TRI* evaluation results. The concentrations of Pb, As, Ni, Zn, and Cu are almost the highest in the Suhe River and Benghe River tributaries, which may be related to the geographical location: there are frequent human activities along these tributaries. The Suhe River and Benghe River are close to residential and transportation areas and are seriously affected by man-made impacts. For example, pollutants discharged from the industrial park in the western part of the study area enter the Suhe River and Benghe River, which is also the reason for the high concentration in the tributaries. The concentration of metals at DN05 remains at a high level, probably because this point is located at the intersection of three tributaries and is greatly affected by tributary pollution. It is worth noting that at sampling point DN23, the concentration of metals is significantly higher than at adjacent sampling points upstream and downstream. DN23 is located downstream of Hexin Island (Yihe North Island), where the water flow is slow and metals are easy to accumulate, and this point may be affected by the Xiaobudong Rubber Dam. Rubber dams can reduce the water flow rate and thus have a blocking effect; the reduced flow rate accelerates the deposition of suspended matter in the

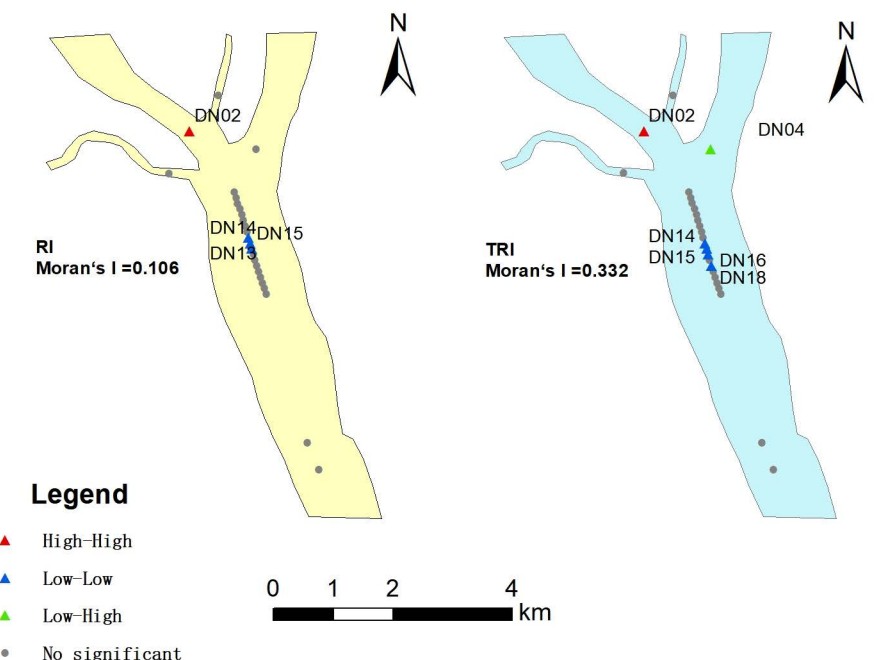

**Fig 6. Spatial autocorrelation results of *RI* and *TRI* in the study area.**

**Table 4. Comparison of metal concentrations in historical sediments of the main and tributary rivers of the Yihe River (mg kg⁻¹).**

| Years | River | As | Cd | Cr | Cu | Hg | Pb |
|---|---|---|---|---|---|---|---|
| 2009 | Benghe River | 5.21 | 0.25 | 74.23 | 34.68 | 0.07 | 37.12 |
| | Suhe River | 7.09 | 0.23 | 70.35 | 32.15 | 0.23 | 36.68 |
| | Liuqinghe River | 5.97 | 1.56 | 84.64 | 50.63 | 0.06 | 59.28 |
| | Main stream of Yihe River | 7.09 | 0.27 | 67.88 | 46.29 | 0.17 | 40.78 |
| 2023 | Benghe River | 5.73 | 0.18 | 69.70 | 29.34 | 0.05 | 28.13 |
| | Suhe River | 7.39 | 0.45 | 67.63 | 37.50 | 0.44 | 45.28 |
| | Liuqinghe River | 2.35 | 0.14 | 33.03 | 14.97 | 0.07 | 24.25 |
| | Main stream of Yihe River | 1.00 | 0.03 | 31.63 | 3.17 | 0.01 | 17.64 |

water. This point is closest to the rubber dam upstream, and the sedimentation effect may be more obvious. In terms of time, compared with the data measured in 2009 (Table 4) [18], the Hg and Cd contents of the Suhe River increased, which may be related to the development and utilization of the North and South Streets of the Suhe River. In comparison, the contents of As, Cu, Cr, Pb, and Cd in the Liuqinghe River decreased significantly. The local government implemented the treatment of the Liuqinghe River in 2017, which may be the reason for this decrease. Meanwhile, the contents of As, Cu, Cr, Pb, Hg, and Cd in the main stream of the Yihe River also decreased significantly. Normally, the metal content in sediments is hundreds of times or even more than that in water bodies. Compared with the metal content in the water bodies of the Yihe River in historical studies [36,37], the metal content in the sediments of the Yihe River is several to 100 times that of the metal content in historical water bodies, indicating that the metal content in the Yihe River has decreased since 2013. The decrease is closely related to the local government's management and protection of the Yihe River.

We compared the metal content found in Yihe River sediments with rivers in other regions analyzed in the literature (Table 5). Compared with the Yishui County area in the upper reaches of the Yihe River, the metal content in Linyi City is significantly lower, which is related to the developed industry and agriculture in Yishui County in the upper reaches of the Yihe River. Statistics show that the comprehensive energy consumption of industries above designated size in Yishui County is 1.6 times that of urban areas, and the use of agricultural fertilizers is 1.25 times that of urban areas, resulting in a higher metal input to the Yishui County area. Comparing the Yihe River with other rivers in Shandong Province, the metal content in the Xiaoqinghe River is higher. This may be closely related to the concentration of industries in the Xiaoqinghe River Basin. Historically, petrochemical, steel, chemical and other enterprises have long been present in the Xiaoqinghe River Basin, and metals were imported from industrial sources. These were a major source of metals in the Xiaoqinghe River [38]. The metal contents in the Yihe River and the Daguhe River are similar, which may be related to the fact that the Yihe River and the Daguhe River may be affected by the same type and degree of agriculture and industries. The Yihe River is a typical river in northern China, and a comparative analysis was conducted between it and the rivers in southern China. The Yihe River has higher metal content than the Xiangxi River in Hubei Province, which may be related to different climatic conditions. The Yihe River has a temperate monsoon climate, with an average annual rainfall of 800~848 mm, while Xiangxi River has a subtropical monsoon climate, with an average annual rainfall of 916~1080 mm [39]. The differences in rainfall may have an impact on metal content in sediments. When precipitation increases, on the one hand, it will dilute the concentration of metals; on the other hand, the hydrodynamic conditions in the river will change, and the metals in the sediment will re-enter the water body, reducing the content of metals in the sediment. The Zn and Cu concentrations in Qinjiang River, Guangxi Province are significantly higher than those in Yihe River, and the contents of other metals are similar to those in the Yihe River. This may be related to the mining of local zinc ore and manganese ore in Guangxi Province. Compared with metals in river sediments in other countries in the world, the metal content in the Yihe River is relatively low, which is closely related to factors such as agricultural or industrial activities and urban development in different countries and regions [44–46].

The correlation between metals may reflect the sources and migration pathways of metal elements [47–49]. Table 6 presents the calculated sediment correlation analysis matrix. The data reveal that, except for As~Cd that has no significant correlation, other metals have a significant positive correlation. Among them, there is a strong correlation between Ni, Cu, Zn,

**Table 5. Comparison of average metal concentrations in Yihe River sediments analyzed in this study and the relevant literature (mg kg⁻¹).**

| River | Region | Metal concentration in sediment (mg kg⁻¹) | | | | | | | | References |
|---|---|---|---|---|---|---|---|---|---|---|
| | | Cr | Ni | Cu | Zn | Cd | Pb | As | Hg | |
| Yihe River | Shandong Province, China | 36.72 | 16.22 | 9.89 | 58.73 | 0.12 | 20.79 | 2.26 | 0.04 | This study |
| Yihe River Upstream Yishui County | Shandong Province, China | 131.79 | 42.29 | 24.29 | 165.59 | 0.27 | 21.18 | 2.13 | 0.66 | [40] |
| Daguhe River | Shandong Province, China | 56.81 | – | 18.42 | 69.01 | 0.083 | 13.51 | 9.84 | – | [41] |
| Xiaoqinghe River | Shandong Province, China | 171.61 | 31.99 | 62.77 | 314.25 | 0.62 | 49.06 | 11.10 | – | [38] |
| Xiangxi River | Hubei Province, China | – | 6.2 | 3.21 | 2.4 | 0.17 | 10.59 | 0.65 | 0.05 | [42] |
| Qinjiang River | Guangxi Province, China | 29.78 | 17.82 | 65.45 | 112.49 | 0.02 | 22.99 | – | – | [43] |
| Batin River | Batin Province, Turkey | 19.0 | 24.7 | 27.0 | 57.5 | – | 14.0 | 4.0 | – | [44] |
| Mae Chaem River | Chiang Mai, Thailand | 45.8 | 24.0 | 21.9 | 60.9 | 0.33 | 27.2 | 32.5 | – | [45] |
| Al-Diwaniyah River | Iraq | 29.92 | 77.36 | 28.72 | 40.11 | 0.28 | 43.8 | 23.31 | – | [46] |

**Table 6. Spearman correlation analysis of metals in sediments of Yihe River.**

|  | Cr | Ni | Cu | Zn | Cd | Pb | As | Hg | pH | SOM |
|---|---|---|---|---|---|---|---|---|---|---|
| Cr | – |  |  |  |  |  |  |  |  |  |
| Ni | 0.786 ** | – |  |  |  |  |  |  |  |  |
| Cu | 0.738 ** | 0.928 ** | – |  |  |  |  |  |  |  |
| Zn | 0.701 ** | 0.952 ** | 0.920 ** | – |  |  |  |  |  |  |
| Cd | 0.506 ** | 0.543 ** | 0.711 ** | 0.572 ** | – |  |  |  |  |  |
| Pb | 0.544 ** | 0.685 ** | 0.770 ** | 0.714 ** | 0.738 ** | – |  |  |  |  |
| As | 0.569 ** | 0.845 ** | 0.778 ** | 0858 ** | 0.369 | 0.462 ** | – |  |  |  |
| Hg | 0.561 ** | 0.667 ** | 0.604 ** | 0.671 ** | 0.635 ** | 0.644 ** | 0.652 ** | – |  |  |
| pH | -0.242 | -0.085 | -0.271 | -0.182 | -0.646 ** | -0.332 | -0.111 | -0.37 | – |  |
| SOM | 0.563 ** | 0.712 ** | 0.745 ** | 0.819 ** | 0.402 * | 0.536 ** | 0.732 ** | 0.537 ** | −0.269 | – |

*Correlation is significant at the 0.05 level (two-tailed).

**Correlation is significant at the 0.01 level (two-tailed).

Cr, and As, and their spatial distributions are similar (Fig 2), indicating that they may come from the same source. Eight metals (metalloids) showed a negative correlation with pH, indicating that increasing pH may reduce the content of metals in sediments. Sediment organic matter (SOM) can affect the ecotoxicity, environmental changes and geochemical status of trace metals in sediments [50]. The organic matter in the Yihe River sediments has a significant positive correlation with eight metals. Among them, the correlation with Zn (0.819) and Cu (0.745) is the highest, indicating that organic matter in the Yihe River sediments has a significant positive correlation with metals. The content or organic matter is closely related to metals concentration, thus they may have the same source or migration pathway.

The distribution similarity of metals in surface sediments was studied using the hierarchical analysis method (HACA). As shown in Fig 7, in general, Cu and Hg were divided into cluster I; the other metals formed cluster II, which may indicate the existence of two different metal sources. Hg is widely used as an industrial additive. Cd is the main raw material in modern industrial production and is widely used in industrial products. It is also produced in the electroplating industry and metal smelting. According to the heavy metal distribution map, Hg and Cd concentrations reached the highest values in the Suhe River, a tributary of the Yihe River, which may be related to the industrial park on the west side of the area. Therefore, cluster I may come from industrial sources. Cluster II is further divided into sub-cluster I with Zn and Pb, and sub-cluster II with As, Cu, Ni, and Cr. Zn is related to vehicle exhaust emissions and tire wear, while Pb may come from vehicle exhaust emissions [51]. Previous studies have shown that Pb is easily combined with atmospheric particles, transported and finally deposited in soils and sediments [52]. As and Cu are usually added to agricultural insecticides or pesticides, so As and Cu may come from agricultural sources. Ni and Cr are generally considered to be natural elements and are affected by natural factors. Therefore, sub-cluster II may come from a mixed source of agricultural and natural sources.

The principal component analysis method was further used for source analysis, and the results were similar to those of the HACA. PCA identified two principal components (PCs), which explained 89.6% of the total variance, indicating that there are different controlling factors or sources of metals in the Yihe River sediments (Fig 8). The metal loads measured in the sediment samples of the Yihe River were classified using the Varimax rotation method. The eigenvalues, explained variance, percentage of variance extracted by different components, and factor loadings of different variables are shown in S6 Table. PC1 accounts for 78.4%

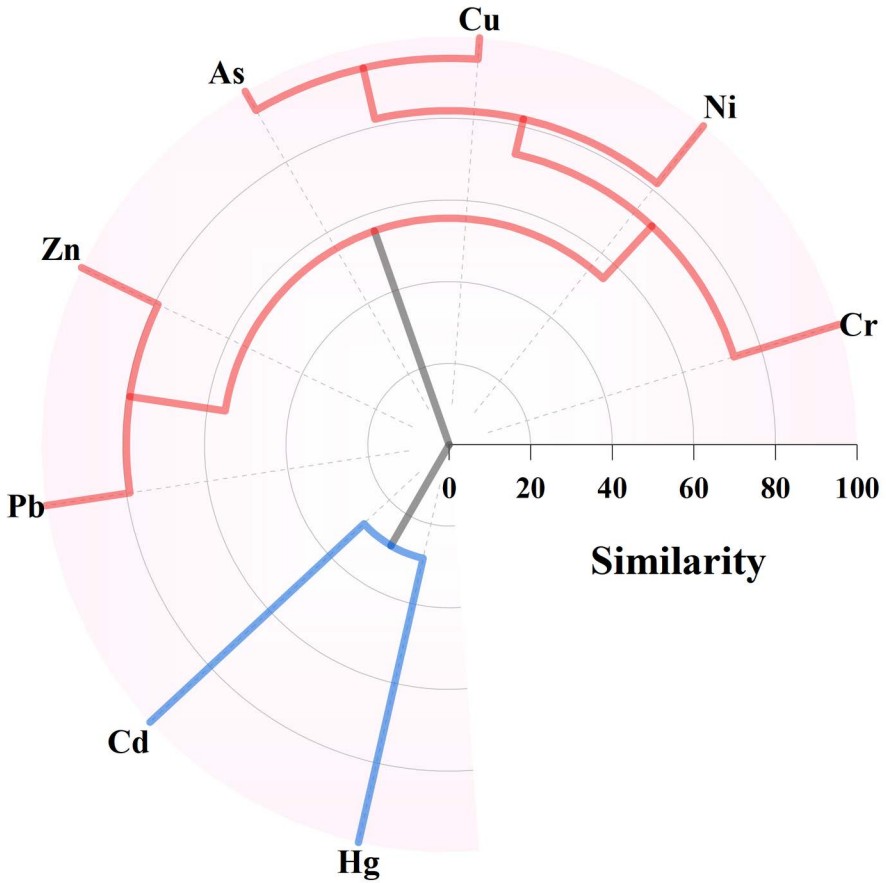

**Fig 7. Dendrogram of HACA based on eight metals concentrations.**

of the total variance, with high loadings for Cr (0.913), Ni (0.901), Cu (0.864), and As (0.906), and medium loadings for Zn (0.73), which is consistent with the high correlation between these metals discussed above. Previous studies have suggested that Cr and Ni are elements of natural origin, mainly affected by natural factors such as the soil parent materials [17,53]. As is an important component in insecticides, herbicides, fungicides, desiccants, defoliants, and animal feed additives, and Cu may originate from the excessive application of pesticides, herbicides or fertilizers in farmland [42,54,55]. The Yihe River Basin is an important area of vegetable production and fruit and flower source for Linyi City. Agricultural activities are frequent along the upper reaches of the Yihe River, and there are dense parks and squares along the river and its tributaries. A large amount of pesticides and herbicides used in agricultural activities are not completely absorbed; the metals contained therein flow into the Yihe River and its tributaries with surface runoff and are deposited in sediments through physical adsorption or chemical precipitation. Therefore, PC1 is a mixed source of agricultural activities and natural sources.

PC2 explains 11.2% of the total variance, of which Cd (0.936) reaches high load, Pb (0.794) and Hg (0.682) are in medium loads. PC2 can be represented as a mixed source of traffic and industrial pollution. Studies have shown that environmental Pb is closely related to the combustion of coal, oil and other fossil fuels, and is generated by paint, shipping and industrial wastewater discharge [6,56–58]. According to the Linyi Statistical Yearbook 2023, the

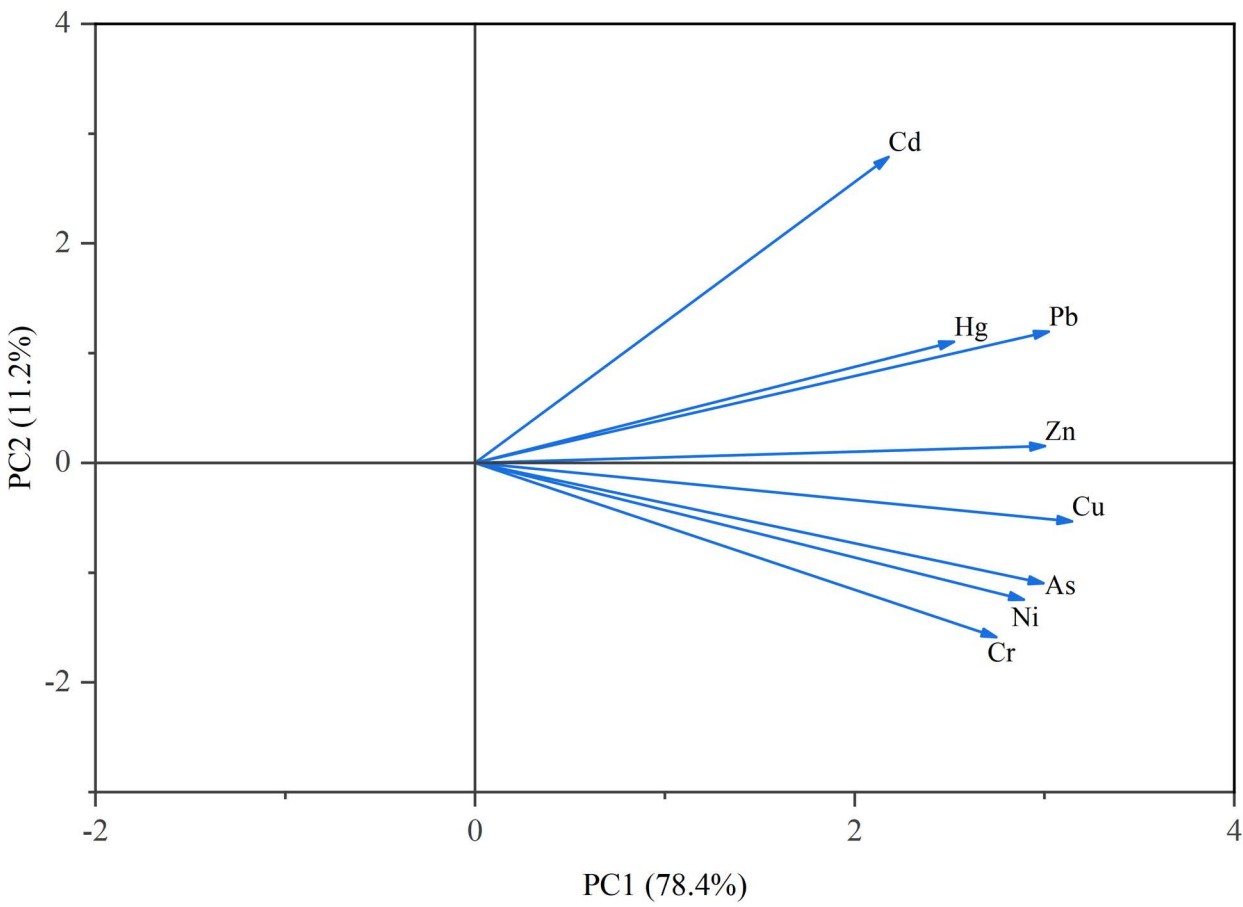

**Fig 8. Principal component analysis of metals in sediments.**

number of civil motor vehicles in Linyi City reached 3,527,516 in 2022, and the traffic along the Yihe River was well developed, with a multitude of cross-river roads and riverside roads. Gasoline residues and automobile exhaust were a major source of Pb in the study area. Hg mainly comes from the combustion of fossil fuels such as coal. In many industrial production processes, Hg is widely used as additives and catalysts [59]. Cd mainly originates from metal smelting and electroplating. In 2022, the total industrial output value of Linyi City accounted for 33.1% of the total GDP, and a total of 20,402,545 tons of standard coal were consumed. Among them, the standard coal consumed by the metal smelting and rolling processing industry and the metal preparation industry accounted for about 40.15% of the total consumption, which may have been the source of Hg and Cd in sediments.

The APCS-MLR method can overcome the weakness of PCA being insufficient to provide quantitative information based on the contribution of each source type. Based on the PCA results, the APCS-MLR receptor model was used to obtain the contribution of metals from various sources in the sediments of the Yihe River, and the results are shown in Table 7. The contribution of PC1 to different metals ranges from 23.49% (Cd) to 88.99% (Cr), and was 84.80% (Ni), 82.92% (As), 77.03% (Cu), and 70.07% (Zn), respectively. PC2 accounts for 76.51% of Cd, 41.56% of Pb, and 43.09% of Hg. 67.83% of metals in sediments came from mixed sources of agricultural activities and natural sources, and the average contribution

**Table 7.  APCS-MLR calculation of metal source contribution rate (%) and predicted ratio (M/P).**

|    | PC1 | PC2 | M/P | $R^2$ |
|----|-----|-----|-----|----|
| Cr | 88.99 | 11.01 | 1.0 | 0.869 |
| Ni | 84.80 | 15.20 | 1.0 | 0.897 |
| Cu | 77.03 | 22.97 | 1.0 | 0.984 |
| Zn | 70.07 | 29.93 | 1.0 | 0.884 |
| Cd | 23.49 | 76.51 | 1.0 | 0.890 |
| Pb | 58.44 | 41.56 | 1.0 | 0.972 |
| As | 82.92 | 17.08 | 1.0 | 0.941 |
| Hg | 56.91 | 43.09 | 1.0 | 0.663 |

percentage of traffic pollution and industrial sources was 32.17%. According to the multiple regression results, the goodness of fit $R^2$ is between 0.663 and 0.984, and the measured value to predicted value (M/P) of all elements are equal to 1, indicating that the constructed APCS-MLR receptor model is suitable for estimating the source distribution of metals in sediment samples from different land use areas in the Yihe River, and the evaluation results are relatively reliable.

In response to the problem of metal pollution in the sediments of the Yihe River, it is crucial to implement effective management and control measures based on actual conditions. We proposed some ideas for the prevention and control of metals in Yihe River sediments. (1) Strengthen the supervision and performance evaluation of the mechanism for eliminating obsolete production capacity, and regularly disclose and update the negative list of enterprises that need to be withdrawn within a time limit. For emerging industries involving metal emissions, implement the strategy of "equal replacement" or "reduction replacement" to control the total amount of emissions [60]. (2) Strengthen the supervision of key polluting industries, especially the environmental governance of high-pollution industries such as plastic processing and electroplating, to ensure that pollution is effectively controlled. In addition, actively promote the circular economy and clean production model to promote efficient resource utilization and environmental protection. (3) In order to reduce the emission of metals in motor vehicle exhaust, it is necessary to formulate and implement strict emission management policies. Vehicles that do not meet emission standards should be gradually restricted and eliminated. Vehicle owners should be encouraged to install exhaust treatment devices to improve emission purification efficiency; and policy incentives such as car purchase subsidies should be used to promote the popularization of electric vehicles; and citizens should be encouraged to take public transportation to effectively reduce the content of metals in atmospheric deposition sources. (4) The application of chemical fertilizers should adhere to scientific and rational principles, aiming to enhance their efficiency in nutrient delivery. It is imperative to boost the utilization ratio of these fertilizers and rigorously ban the usage of any fertilizer that contains excessively high levels of metals. Reuse crop straw, livestock and poultry manure and other agricultural waste into organic fertilizer or biomass energy, reducing the use of chemical fertilizers and metal pollution. The above measures can effectively solve the problem of metal pollution in the sediments of the Yihe River.

## Conclusion

The average concentrations of eight metals in the sediments of the Linyi urban section of the Yihe River were Cr (36.72 mg kg$^{-1}$), Ni (16.22 mg kg$^{-1}$), Cu (9.89 mg kg$^{-1}$), and Zn (58.73 mg kg$^{-1}$), Cd (0.12 mg kg$^{-1}$), Pb (20.79 mg kg$^{-1}$), As (2.26 mg kg$^{-1}$) and Hg (0.04 mg kg$^{-1}$).

The spatial distribution of metals mainly shows that the tributaries Suhe River (DN01), Benghe River (DN02), and Liuqinghe River (DN03) are seriously polluted. Metals also exhibit higher concentrations in the sediments at the intersection of the three tributaries and the North Island of Yihe River.

Metals in the sediments of the tributaries of Suhe River, Benghe River, Liuqinghe River and the North Island of Yihe River have low toxicity risk and strong ecological risk. Metals are in a pollution-free state in most areas of the main stream of Yihe River, with low ecological risk and no toxicological risk. Hg and Cd are the main contributors to the ecological risks.

Metals in the sediments of the Linyi urban section of the Yihe River mainly come from two aspects: Cr, Ni, Cu, Zn, and As mainly originate from natural sources and agricultural activities, with a contribution rate of 67.83%; Cd, Pb and Hg are mainly produced by transportation and industry, with an average proportion of 32.17%. These research results provide an important reference for pollution control and planning prevention in the Yihe River Basin.

## Supporting information

**S1 Table. The distribution of specific metals.**
(DOCX)

**S2 Table. *PLI* values calculated at each sampling point.**
(DOCX)

**S3 Table. *$I_{geo}$* values calculated at each sampling point.**
(DOCX)

**S4 Table. *RI* values calculated at each sampling point.**
(DOCX)

**S5 Table. *TRI* values calculated at each sampling point.**
(DOCX)

**S6 Table. Characteristic values and cumulative contribution rates of metals in sediments.**
(DOCX)

## Acknowledgments

The authors sincerely thank all the staff for their diligent assistance in the sampling process. Their unwavering support throughout the sampling phase have been crucial for the quality and integrity of our research. And we would like to express our sincere appreciation to The Seventh Geological Bridge of Shandong Provincial Bureau of Geology and Mineral for providing us with excellent experimental conditions. Their facilities and resources have been invaluable in conducting our analyses and obtaining reliable results.

## Author contributions

**Conceptualization:** Qinghai Deng, Liping Zhang.

**Funding acquisition:** Qinghai Deng.

**Investigation:** Fuquan Li, Lei Guo, Dan Shi.

**Methodology:** Qinghai Deng, Guizong Sun.

**Visualization:** Zhenzhou Sun, Jingjing Yang.

**Writing – original draft:** Guizong Sun.

**Writing – review & editing:** Qinghai Deng, Liping Zhang.

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
