## [Decision Letter · Decision Letter 0]

14 Aug 2024

PONE-D-24-25973Assessment and source analysis of heavy metal pollution in sediments from the urban section of Yihe River, Linyi City, Shandong, ChinaPLOS ONE

Dear Dr. Zhang,

Thank you for submitting your manuscript to PLOS ONE. After careful consideration, we feel that it has merit but does not fully meet PLOS ONE’s publication criteria as it currently stands. Therefore, we invite you to submit a revised version of the manuscript that addresses the points raised during the review process.

 Please submit your revised manuscript by Sep 28 2024 11:59PM. If you will need more time than this to complete your revisions, please reply to this message or contact the journal office at plosone@plos.org . Please include the following items when submitting your revised manuscript:

We look forward to receiving your revised manuscript.

Kind regards,

Timothy Omara, PhD

Academic Editor

PLOS ONE

Journal Requirements:

1. When submitting your revision, we need you to address these additional requirements. Please ensure that your manuscript meets PLOS ONE's style requirements, including those for file naming. The PLOS ONE style templates can be found at https://journals.plos.org/plosone/s/file?id=wjVg/PLOSOne_formatting_sample_main_body.pdf and https://journals.plos.org/plosone/s/file?id=ba62/PLOSOne_formatting_sample_title_authors_affiliations.pdf 2. In your Methods section, please provide additional information regarding the permits you obtained for the work. Please ensure you have included the full name of the authority that approved the field site access and, if no permits were required, a brief statement explaining why.  3. Thank you for stating in your Funding Statement: This research was financially supported by the Shandong Provincial Natural Science Foundation (Grant No. ZR2022MD032). The funding agency had no role in study design, data collection and analysis, decision to publish, or preparation of the manuscript. The authors are grateful for the support., Please provide an amended statement that declares all the funding or sources of support (whether external or internal to your organization) received during this study, as detailed online in our guide for authors at http://journals.plos.org/plosone/s/submit-now.  Please also include the statement “There was no additional external funding received for this study.” in your updated Funding Statement. Please include your amended Funding Statement within your cover letter. We will change the online submission form on your behalf. 4. We note that your Data Availability Statement is currently as follows: All relevant data are within the manuscript and its Supporting Information files. Please confirm at this time whether or not your submission contains all raw data required to replicate the results of your study. Authors must share the “minimal data set” for their submission. PLOS defines the minimal data set to consist of the data required to replicate all study findings reported in the article, as well as related metadata and methods (https://journals.plos.org/plosone/s/data-availability#loc-minimal-data-set-definition). For example, authors should submit the following data: - The values behind the means, standard deviations and other measures reported;- The values used to build graphs;- The points extracted from images for analysis. Authors do not need to submit their entire data set if only a portion of the data was used in the reported study. If your submission does not contain these data, please either upload them as Supporting Information files or deposit them to a stable, public repository and provide us with the relevant URLs, DOIs, or accession numbers. For a list of recommended repositories, please see https://journals.plos.org/plosone/s/recommended-repositories. If there are ethical or legal restrictions on sharing a de-identified data set, please explain them in detail (e.g., data contain potentially sensitive information, data are owned by a third-party organization, etc.) and who has imposed them (e.g., an ethics committee). Please also provide contact information for a data access committee, ethics committee, or other institutional body to which data requests may be sent. If data are owned by a third party, please indicate how others may request data access. 5. PLOS requires an ORCID iD for the corresponding author in Editorial Manager on papers submitted after December 6th, 2016. Please ensure that you have an ORCID iD and that it is validated in Editorial Manager. To do this, go to ‘Update my Information’ (in the upper left-hand corner of the main menu), and click on the Fetch/Validate link next to the ORCID field. This will take you to the ORCID site and allow you to create a new iD or authenticate a pre-existing iD in Editorial Manager. Please see the following video for instructions on linking an ORCID iD to your Editorial Manager account: https://www.youtube.com/watch?v=_xcclfuvtxQ 6. We note that Figure 1 in your submission contain map images which may be copyrighted. All PLOS content is published under the Creative Commons Attribution License (CC BY 4.0), which means that the manuscript, images, and Supporting Information files will be freely available online, and any third party is permitted to access, download, copy, distribute, and use these materials in any way, even commercially, with proper attribution. For these reasons, we cannot publish previously copyrighted maps or satellite images created using proprietary data, such as Google software (Google Maps, Street View, and Earth). For more information, see our copyright guidelines: http://journals.plos.org/plosone/s/licenses-and-copyright. We require you to either present written permission from the copyright holder to publish these figures specifically under the CC BY 4.0 license, or remove the figures from your submission: a. You may seek permission from the original copyright holder of Figure 1 to publish the content specifically under the CC BY 4.0 license.   We recommend that you contact the original copyright holder with the Content Permission Form (http://journals.plos.org/plosone/s/file?id=7c09/content-permission-form.pdf) and the following text:“I request permission for the open-access journal PLOS ONE to publish XXX under the Creative Commons Attribution License (CCAL) CC BY 4.0 (http://creativecommons.org/licenses/by/4.0/). Please be aware that this license allows unrestricted use and distribution, even commercially, by third parties. Please reply and provide explicit written permission to publish XXX under a CC BY license and complete the attached form.” Please upload the completed Content Permission Form or other proof of granted permissions as an "Other" file with your submission. In the figure caption of the copyrighted figure, please include the following text: “Reprinted from [ref] under a CC BY license, with permission from [name of publisher], original copyright [original copyright year].” b. If you are unable to obtain permission from the original copyright holder to publish these figures under the CC BY 4.0 license or if the copyright holder’s requirements are incompatible with the CC BY 4.0 license, please either i) remove the figure or ii) supply a replacement figure that complies with the CC BY 4.0 license. Please check copyright information on all replacement figures and update the figure caption with source information. If applicable, please specify in the figure caption text when a figure is similar but not identical to the original image and is therefore for illustrative purposes only.The following resources for replacing copyrighted map figures may be helpful: USGS National Map Viewer (public domain): http://viewer.nationalmap.gov/viewer/The Gateway to Astronaut Photography of Earth (public domain): http://eol.jsc.nasa.gov/sseop/clickmap/Maps at the CIA (public domain): https://www.cia.gov/library/publications/the-world-factbook/index.html and https://www.cia.gov/library/publications/cia-maps-publications/index.htmlNASA Earth Observatory (public domain): http://earthobservatory.nasa.gov/Landsat: http://landsat.visibleearth.nasa.gov/USGS EROS (Earth Resources Observatory and Science (EROS) Center) (public domain): http://eros.usgs.gov/#Natural Earth (public domain): http://www.naturalearthdata.com/

Reviewers' comments:

Reviewer's Responses to Questions

**Comments to the Author**

1. Is the manuscript technically sound, and do the data support the conclusions?

Reviewer #1: No

Reviewer #2: Yes

2. Has the statistical analysis been performed appropriately and rigorously? 

Reviewer #1: No

Reviewer #2: Yes

3. Have the authors made all data underlying the findings in their manuscript fully available?

Reviewer #1: Yes

Reviewer #2: Yes

4. Is the manuscript presented in an intelligible fashion and written in standard English?

Reviewer #1: No

Reviewer #2: Yes

5. Review Comments to the Author

**Reviewer #1: ** Through the review of the manuscript, it is considered that there are three problems, so it is not recommended to publish.

1. Scientific value of research topics

Although the manuscript focuses on the risk assessment of heavy metal pollution in the Linyi section of the Yihe River, which is of great significance to environmental protection and water resources management, the scope of the study is relatively limited, focusing only on heavy metal pollution in sediments, without considering the migration and transformation of heavy metals in water and their impact on aquatic ecosystems. In addition, the assessment and risk assessment of heavy metal pollution usually require long-term monitoring and comprehensive analysis, while the short-term and single-point sampling in this study may not be enough to fully reflect the spatial and temporal distribution characteristics of heavy metal pollution. Therefore, the scientific value of research topics is limited to a certain extent.

2. Innovation of research methods

In terms of research methods, although pollution load index, geological accumulation index and other methods are used to evaluate heavy metal pollution, these methods are relatively traditional and do not show significant innovation in research methods. Especially in the part of source apportionment, the methods of correlation analysis and absolute principal components-multiple linear regression (APCs-MLR) can identify the sources of heavy metals, but these methods have been widely used in the field of environmental science, which do not reflect the uniqueness and innovation of research methods.

3. The practicability of the research results

Although the research results show that there is no heavy metal pollution in most areas of the main stream of yi river, and the ecological risk is low, the ecological risk of tributaries and central islands is high. However, these research results have limited practical guiding significance for local environmental management departments and relevant state departments. On the one hand, the research results did not put forward specific pollution control and treatment measures; On the other hand, the research results do not provide effective policy suggestions for the control and governance of pollution sources. Therefore, the research results are not practical enough to be directly applied to environmental management and policy formulation.

**Reviewer #2:**  Title: Heavy metal pollution in sediments from the urban section of Yihe River, Linyi City, Shandong, China

The team should adopt a chemist to execute risk assessment, adding value to the publication.

Abstract

Measurement of potentially toxic substances i.e. heavy, metals levels in sediment differs from what is in the water levels. Comparisons are crucial for discussion towards health

Introduction

Since this is an ecological risk assessment, the three phases:

• Phase 1 - Problem Formulation Information was documented.

• Phase 2 - Analysis was also well documented

• Phase 3 - Risk Characterization: this was not well discussed and the target populations should be clearly articulated. Use Hazard index and reference doses for vulnerable populations.

The above will enrich the introduction, methods and discussion.

Enrich discussions by comparing sediment samples with airborne and water samples and why sediment are crucial source.

Methods

These are detailed and well-written.

Limitations to methods and samples are missing. Sediment samples are usually not representative of water sample amounts of pollutants even though Toxicology assessments document risks for exposure through water uptake.

Results

The distribution of HMs in surface sediments should be presented in tables and possibly also show the max allowable limits by various associations.

The spatial maps show individual pollutant distribution so that the heavily polluted sections of the river are displayed, linked to sources of contamination and strategies of how to control can be displayed.

Risk assessment can be displayed using probability index, global rand scatter plots and Local Moran's I clustering plots

Discussions

You discussed sources, distribution and unfortunately the health burden was not clearly articulated.

Conclusions

Information on health burden not presented.

6. PLOS authors have the option to publish the peer review history of their article (what does this mean? ). If published, this will include your full peer review and any attached files.

**Do you want your identity to be public for this peer review?** For information about this choice, including consent withdrawal, please see our Privacy Policy .

Reviewer #1: No

Reviewer #2: **Yes: ** DR TAMALE ANDREW PhD

---

## [Author Response · Author response to Decision Letter 1]

7 Nov 2024

Response to Reviewers

Dear Editor and Reviewers:

Thank you for your letter and for the reviewers' comments concerning our manuscript entitled “Assessment and source analysis of heavy metal pollution in sediments from the urban section of Yihe River, Linyi City, Shandong, China” (Manuscript Number: PONE-D-24-25973). Those comments are all valuable and very helpful for revising and improving our paper, as well as the important guiding significance to our researches. We have studied all comments carefully and have made correction that we hope meet with approval. The revised sections have been distinctly highlighted in red within the paper. The main corrections in the paper and the responds to the reviewers' comments are as following:

Responds to the reviewer' s comments:

Reviewer 1:

Question 1. Scientific value of research topics

Although the manuscript focuses on the risk assessment of heavy metal pollution in the Linyi section of the Yihe River, which is of great significance to environmental protection and water resources management, the scope of the study is relatively limited, focusing only on heavy metal pollution in sediments, without considering the migration and transformation of heavy metals in water and their impact on aquatic ecosystems. In addition, the assessment and risk assessment of heavy metal pollution usually require long-term monitoring and comprehensive analysis, while the short-term and single-point sampling in this study may not be enough to fully reflect the spatial and temporal distribution characteristics of heavy metal pollution. Therefore, the scientific value of research topics is limited to a certain extent.

Response: Thank you very much for your valuable comments. This study mainly focuses on the risk assessment and source analysis of heavy metal pollution in the Linyi section of the Yihe River. Although the scope is limited, it is of great significance for regional environmental protection and water resource management. In addition, our current data is limited, but we will continue to monitor and analyze heavy metal pollution in the area in the future.

Question 2. Innovation of research methods

In terms of research methods, although pollution load index, geological accumulation index and other methods are used to evaluate heavy metal pollution, these methods are relatively traditional and do not show significant innovation in research methods. Especially in the part of source apportionment, the methods of correlation analysis and absolute principal components-multiple linear regression (APCs-MLR) can identify the sources of heavy metals, but these methods have been widely used in the field of environmental science, which do not reflect the uniqueness and innovation of research methods.

Response: Thank you very much for your comments. Although our research methods are relatively traditional and widely used, they have also been proven to be effective for heavy metal pollution assessment and source apportionment. Of course, we will strive to develop some new models and evaluation methods in the future.

Question 3. The practicability of the research results

Although the research results show that there is no heavy metal pollution in most areas of the main stream of yi river, and the ecological risk is low, the ecological risk of tributaries and central islands is high. However, these research results have limited practical guiding significance for local environmental management departments and relevant state departments. On the one hand, the research results did not put forward specific pollution control and treatment measures; On the other hand, the research results do not provide effective policy suggestions for the control and governance of pollution sources. Therefore, the research results are not practical enough to be directly applied to environmental management and policy formulation.

Response: Thank you very much for your comments. We have deeply understood and attached great importance to the shortcomings of the research you pointed out in proposing specific pollution control measures and policy recommendations. In response to your feedback, we have made corresponding supplements and improvements in the revised manuscript. For example, strengthening the supervision and performance evaluation of the mechanism for phasing out outdated production capacity, enhancing environmental governance supervision of key polluting industries, formulating and implementing strict emission management policies, reducing heavy metal emissions from motor vehicle exhaust, prohibiting the use of fertilizers with excessive heavy metals, and so on. These contents are in the last paragraph of the discussion section and have been highlighted in red font.

Reviewer 2:

Question 1. Abstract

Measurement of potentially toxic substances i.e. heavy, metals levels in sediment differs from what is in the water levels. Comparisons are crucial for discussion towards health.

Response: Thank you for your feedback. We have considered your suggestions and supplemented the significance of studying heavy metals in sediments and added a human health risk assessment model to assess the health risks of heavy metals in sediments for local residents. The revisions are in the first paragraph of the Abstract section.

Question 2. Introduction

Since this is an ecological risk assessment, the three phases:

Phase 1 - Problem Formulation Information was documented.

Phase 2 - Analysis was also well documented

Phase 3 - Risk Characterization: this was not well discussed and the target populations should be clearly articulated. Use Hazard index and reference doses for vulnerable populations.

The above will enrich the introduction, methods and discussion.

Enrich discussions by comparing sediment samples with airborne and water samples and why sediment are crucial source.

Response: Thank you for your recognition and affirmation of our work. We have considered your suggestion and added the health risks of heavy metals to the target population (local residents) and discussed the vulnerable populations (children). We also added the reasons why sediment is a crucial source. The revisions are in the first paragraph of the Introduction section.

Question 3.Methods

These are detailed and well-written.

Limitations to methods and samples are missing. Sediment samples are usually not representative of water sample amounts of pollutants even though Toxicology assessments document risks for exposure through water uptake.

Response: Thank you very much for your careful review of our study and your valuable comments. We acknowledge that there is indeed a complex relationship between the pollutant content in sediments and the content in water bodies, and they cannot be simply equated or represented. Sediments are both "sinks" and "sources" of pollutants, and their pollutant concentrations may be affected by many factors. However, we are sorry that in this study, we do not have data on heavy metals in water bodies, so we cannot add them to the manuscript. We added the explanation in the fourth paragraph of the Sample collection and analysis section of the Materials and Methods chapter of the manuscript that no water sample data were collected.

Question 4. Results

The distribution of HMs in surface sediments should be presented in tables and possibly also show the max allowable limits by various associations.

The spatial maps show individual pollutant distribution so that the heavily polluted sections of the river are displayed, linked to sources of contamination and strategies of how to control can be displayed.

Risk assessment can be displayed using probability index, global rand scatter plots and Local Moran's I clustering plots.

Response: Thank you for your valuable feedback. We have taken your suggestion into consideration and made the following changes to the manuscript. We included the heavy metal concentrations at each sampling point (Table S2) in the supplementary material. We have updated other association max allowable limits (Table 3). And we incorporate probability distribution into risk assessment (Fig 3b) and supplement Moran’s cluster diagrams for ecological and toxic risks (Fig 6). Revisions have been highlighted in red in the manuscript.

Question 5. Discussions

You discussed sources, distribution and unfortunately the health burden was not clearly articulated.

Response: Thank you very much for your thorough review and valuable comments. We have fully considered your suggestion and added the discussion on health risks to the fourth paragraph of the Discussion section.

Question 6. Conclusion

Information on health burden not presented.

Response: Thank you for your valuable feedback. We have taken your suggestion into consideration and added the results on health burden in the fourth paragraph of the Conclusion section.

We have carefully reviewed the comments provided by the reviewers and made the necessary modifications and explanations. We have diligently worked to enhance the manuscript by implementing specific changes. These alterations do not affect the content or structure of the paper.

We sincerely appreciate the diligent efforts of Editors and Reviewers, and we hope that the revision will meet with approval.

Once again, thank you very much for your comments and suggestions.

Yours sincerely,

Liping Zhang

---

## [Decision Letter · Decision Letter 1]

22 Dec 2024

PONE-D-24-25973R1Assessment and source analysis of heavy metal pollution in sediments from the urban section of Yihe River, Linyi City, Shandong, ChinaPLOS ONE

Dear Dr. Zhang,

Thank you for submitting your manuscript to PLOS ONE. After careful consideration, we feel that it has merit but does not fully meet PLOS ONE’s publication criteria as it currently stands. Therefore, we invite you to submit a revised version of the manuscript that addresses the points raised during the review process.

We look forward to receiving your revised manuscript.

Kind regards,

Timothy Omara

Academic Editor

PLOS ONE

Additional Editor Comments :

Dear Authors,

The reviewers have now re-evaluated your revised manuscript, and as you will see, there are still significant concerns that need to be addressed. Additionally, I have attached the MS Word version of the manuscript with some suggestions for your consideration. Please address the following points:

1. The Discussion section in its current form is too regionally focused. Most of the studies cited are exclusively from China, which limits the broader applicability and significance of your findings. Please compare your results with studies from other parts of the world to strengthen this section.

2. Compare your results of heavy metals in sediments with benchmarks such as the average shale values, toxicity reference values, and consensus-based sediment quality guidelines. Using the “China Soil Standards – Farmland” is not appropriate in this context since these are sediments from a river system.

3. PCA results could be better presented graphically, for example, using biplots, as this is a best practice for clarity and accessibility. If necessary, the detailed results currently shown in Table 8 can be moved to the supplementary files. Graphical presentation makes PCA loadings much easier to interpret.

4. Hierarchical Cluster Analysis (HCA) could provide additional insights into the data. Consider incorporating it for a more comprehensive analysis.

Reviewers' comments:

Reviewer's Responses to Questions

**Comments to the Author**

1. If the authors have adequately addressed your comments raised in a previous round of review and you feel that this manuscript is now acceptable for publication, you may indicate that here to bypass the “Comments to the Author” section, enter your conflict of interest statement in the “Confidential to Editor” section, and submit your "Accept" recommendation.

Reviewer #1: (No Response)

Reviewer #3: (No Response)

2. Is the manuscript technically sound, and do the data support the conclusions?

Reviewer #1: No

Reviewer #3: Yes

3. Has the statistical analysis been performed appropriately and rigorously? 

Reviewer #1: No

Reviewer #3: Yes

4. Have the authors made all data underlying the findings in their manuscript fully available?

Reviewer #1: No

Reviewer #3: Yes

5. Is the manuscript presented in an intelligible fashion and written in standard English?

Reviewer #1: Yes

Reviewer #3: Yes

6. Review Comments to the Author

Reviewer #1: After review, this manuscript is still not ready for publication.

Design Logic. Given that the authors state, "Due to research design and resource constraints, this study did not collect water samples for testing. Therefore, we cannot directly analyze the relationships and differences between heavy metals in water and sediments. However, through detailed sediment analysis, we still revealed some important characteristics of heavy metal contamination in the area." From the perspective of scientific rigor, the authors' research design is inherently flawed, with subsequent conclusions primarily derived from inference rather than actual experimental data.

Data Quality. The authors emphasize quality control in data analysis but neglect repetitive testing of samples. As a result, the statistical stability and variability of the data cannot be confirmed.

The graphical materials provided by the authors are of extremely low resolution, making it impossible to clearly understand the data acquisition information.

In summary, this article has issues in terms of design, innovation, and data quality, and is not recommended for publication.

Reviewer #3: Remove all instances of HEAVY metals and alter to metals and metalloids, according to IUPAC standards

Abstract; "The HMs were evaluated by measuring the pollution load index, geoaccumulation index, etc. " etc is not adequate, rewrite

Overall comment: Do not begin sentences with abbreviations, such as HM, alter.

Correct: MATERIAL and methods, no S

How were the sediments sampled, at what depth?

Correct "detection limits" to "limits of detection".

Alter units to IUPAC standards, not mg/kg, but instead, mg kg-1.

Remove the Human Health Risk assessment, humans do not ingest this sediment. Or at least, maintain only dermal contact.

Statistical ANALYSES, plural, correct.

Indicate how data normality was checked in order to employ Pearson correlation.

7. PLOS authors have the option to publish the peer review history of their article (what does this mean? ). If published, this will include your full peer review and any attached files.

**Do you want your identity to be public for this peer review?** For information about this choice, including consent withdrawal, please see our Privacy Policy .

Reviewer #1: No

Reviewer #3: No

---

## [Author Response · Author response to Decision Letter 2]

4 Jan 2025

Response to Reviewers

Dear Editor and Reviewers:

Thank you for your letter and for the reviewers' comments concerning our manuscript entitled “Assessment and source analysis of heavy metal pollution in sediments from the urban section of Yihe River, Linyi City, Shandong, China” (Manuscript Number: PONE-D-24-25973R1). Those comments are all valuable and very helpful for revising and improving our paper, as well as the important guiding significance to our researches. We have studied all comments carefully and have made correction that we hope meet with approval. The revised sections have been distinctly highlighted in red within the paper. The main corrections in the paper and the responds to the reviewers' comments are as following:

Responds to the editor' s comments:

Question 1. The Discussion section in its current form is too regionally focused. Most of the studies cited are exclusively from China, which limits the broader applicability and significance of your findings. Please compare your results with studies from other parts of the world to strengthen this section.

Response: Thank you very much for your valuable comments. We have added studies from world regions for comparison in the manuscript, and we firmly believe that this will make the findings more broadly applicable and meaningful.

Question 2. Compare your results of heavy metals in sediments with benchmarks such as the average shale values, toxicity reference values, and consensus-based sediment quality guidelines. Using the “China Soil Standards – Farmland” is not appropriate in this context since these are sediments from a river system.

Response: Thank you for your valuable comments. We have deleted the part about “China Soil Standards - Farmland” in the manuscript and added the average shale value as the evaluation standard. Since the toxicity reference values are shown in the Risk Assessment Methods section of the manuscript, the toxicity reference valuesare not added here.

Question 3. PCA results could be better presented graphically, for example, using biplots, as this is a best practice for clarity and accessibility. If necessary, the detailed results currently shown in Table 8 can be moved to the supplementary files. Graphical presentation makes PCA loadings much easier to interpret.

Response: Thank you very much for your comments. We have added the PCA biplots to the manuscript and moved the table with eigenvalues and contributions to the supplementary files.

Question 4. Hierarchical Cluster Analysis (HCA) could provide additional insights into the data. Consider incorporating it for a more comprehensive analysis.

Response: Thank you very much for your valuable comments. We have carefully considered your suggestion and combined it with the HACA Dendrogram (Fig 7) for data analysis.

Responds to the reviewer' s comments:

Reviewer 1:

Question 1. Design Logic. Given that the authors state, "Due to research design and resource constraints, this study did not collect water samples for testing. Therefore, we cannot directly analyze the relationships and differences between heavy metals in water and sediments. However, through detailed sediment analysis, we still revealed some important characteristics of heavy metal contamination in the area." From the perspective of scientific rigor, the authors' research design is inherently flawed, with subsequent conclusions primarily derived from inference rather than actual experimental data.

Response: Thank you for your comments. We admit that there may be a logical problem. Due to funding reasons, we did not collect water samples in the early design. We read a lot of literature, summarized and sorted out the experience of predecessors, and combined with the actual situation of Yihe and Linyi City, then we conducted a reasonable analysis.

Question 2. Data Quality. The authors emphasize quality control in data analysis but neglect repetitive testing of samples. As a result, the statistical stability and variability of the data cannot be confirmed.

Response: Thank you for your comments. We apologize for not making it clear in the manuscript. When collecting samples, we collected three samples from the area around each sampling point and mixed them evenly before measuring the obtained sampling data. We have added these details to the manuscript.

Question 3. The graphical materials provided by the authors are of extremely low resolution, making it impossible to clearly understand the data acquisition information.

Response: Thank you for your valuable comments. This is not intentional on our part, but because before uploading the images, they need to be processed by the image processing website designated by the journal, which may reduce the image resolution.

Reviewer 3:

Question 1. Remove all instances of HEAVY metals and alter to metals and metalloids, according to IUPAC standards.

Response: Thank you for your valuable comments. We have changed all "heavy metals " in the manuscript to "metals or metalloids ".

Question 2. Abstract; "The HMs were evaluated by measuring the pollution load index, geoaccumulation index, etc. " etc is not adequate, rewrite. Overall comment: Do not begin sentences with abbreviations, such as HM, alter.

Response: Thank you very much for your comments. We rewrote the methods in the Abstract section and changed all abbreviations in the manuscript to full names.

Question 3. Correct: MATERIAL and methods, no S

Response: Thank you very much for your comments. We have corrected this part.

Question 4. How were the sediments sampled, at what depth?

Response: Thanks for your comments. The Sample collection and analysis section of the manuscript has already described how to collect samples and the sampling depth.

Question 5. Correct "detection limits" to "limits of detection".

Response: Thank you very much for your valuable comments. We have corrected this in the manuscript.

Question 6. Alter units to IUPAC standards, not mg/kg, but instead, mg kg-1.

Response: Thank you very much for your comments. We have corrected these contents in the manuscript and marked them in red font.

Question 7. Remove the Human Health Risk assessment, humans do not ingest this sediment. Or at least, maintain only dermal contact.

Response: Thank you for your valuable feedback. We have taken your suggestion into consideration and deleted the content about human health risk assessment.

Question 8. Indicate how data normality was checked in order to employ Pearson correlation.

Response: Thank you for your valuable comments. We are sorry that we overlooked the conditions for using person correlation analysis before using it. After verification, the data did not conform to the normal distribution well, so person correlation analysis could not be used. We changed it to spearman correlation analysis in the manuscript. Thank you again for your suggestions, which helped us avoid the mistake.

Responds to the manuscript' s comments:

Question 1. Consider revising this title as suggested. It is too long and redundant.

Response: Thank you for your valuable feedback. We have taken your suggestion into consideration and shortened the title.

Question 2. Indicate the analytical technique used for the measurement.

Response: Thank you for your valuable comments. We have added the analytical technique used for the measurement in the Abstract section.

Question 3. Please calculate the enrichment factor (EF) as well. There are already of studies from China, and you should be able to have the Reference metals with established concentrations.

Response: Thank you for your valuable comments. We are sorry that this method cannot be used because the actual values of the reference elements required for the enrichment factor method were not determined during the preliminary design and measurement process.

Question 4. Which software was used for this analysis? Please indicate the version, developer of the software and their location.

Response: Thank you for your valuable comments. We have added information about the software, version and location used in the analysis in the manuscript.

Question 5. About some grammatical errors or minor modifications.

Response: Thank you for your valuable comments. We have corrected these contents in the manuscript.

We have carefully reviewed the comments provided by the reviewers and made the necessary modifications and explanations. We have diligently worked to enhance the manuscript by implementing specific changes. These alterations do not affect the content or structure of the paper.

We sincerely appreciate the diligent efforts of Editors and Reviewers, and we hope that the revision will meet with approval.

Once again, thank you very much for your comments and suggestions.

Yours sincerely,

Liping Zhang

---

## [Editor Report · Decision Letter 2]

13 Jan 2025

PONE-D-24-25973R2Assessment of heavy metal pollution in sediments from the urban section of Yihe River, Linyi City, ChinaPLOS ONE

Dear Dr. Zhang,

Thank you for submitting your manuscript to PLOS ONE. After careful consideration, we feel that it has merit but does not fully meet PLOS ONE’s publication criteria as it currently stands. Therefore, we invite you to submit a revised version of the manuscript that addresses the points raised during the review process.

**ACADEMIC EDITOR: **

Dear Authors,

I have reviewed your revised manuscript and would like to commend you on the significant improvements made since the previous version. However, there are still a few aspects that require further attention to meet the journal's standards:

**1. Clarification on Normality Testing:**

The reviewer comment, “Indicate how data normality was checked in order to employ Pearson correlation,” was not sufficiently addressed. Please clearly specify in the manuscript which normality test was used (e.g., Shapiro-Wilk, Kolmogorov-Smirnov). Use of an inappropriate normality test can lead to false assumptions about the data distribution and, subsequently, the application of incorrect statistical methods.

**2. Steps for Spearman Correlation Analysis:**

Spearman correlation is a non-parametric method that evaluates monotonic relationships by ranking the data. Please indicate the exact steps undertaken to implement it in your analysis. For instance, note any rank transformations or adjustments for tied ranks, as these are required for this method.

**3. Expanded Explanation for HACA Results:**

Please elaborate further on the HACA (Hierarchical Cluster Analysis) results on page 24, sentence after Table 6. Specifically:

(a) **Primary Clusters (broad groups):** Discuss how similar sources (e.g., Cu and Hg from industrial activities) or comparable environmental behavior and mobilization processes (e.g., Pb, Ni, Zn, and Cr adsorbing onto organic matter or clay minerals) could explain these groupings.

(b) **Sub-Clusters (Finer Groupings):** Explain how these may reflect either geochemical similarities (e.g., Cr and Ni) or distinct combinations of anthropogenic and natural contributions (e.g., Cd and Pb arising from industrial emissions with varying inputs from transportation).

(c) Please revise the STATISTICAL ANALYSIS section to indicate that HACA was performed.

We look forward to receiving your revised manuscript.

Kind regards,

Timothy Omara

Academic Editor

PLOS ONE
---

## [Author Response · Author response to Decision Letter 3]

17 Jan 2025

Response to Reviewers

Dear Editor:

Thank you for your letter and for the reviewers' comments concerning our manuscript entitled “Assessment of heavy metal pollution in sediments from the urban section of Yihe River, Linyi City, China” (Manuscript Number: PONE-D-24-25973R2). Those comments are all valuable and very helpful for revising and improving our paper, as well as the important guiding significance to our researches. We have studied all comments carefully and have made correction that we hope meet with approval. The revised sections have been distinctly highlighted in red within the paper. The main corrections in the paper and the responds to the reviewers' comments are as following:

Responds to the Editor' s comments:

Question 1. Clarification on Normality Testing.

The reviewer comment, “Indicate how data normality was checked in order to employ Pearson correlation,” was not sufficiently addressed. Please clearly specify in the manuscript which normality test was used (e.g., Shapiro-Wilk, Kolmogorov-Smirnov). Use of an inappropriate normality test can lead to false assumptions about the data distribution and, subsequently, the application of incorrect statistical methods.

Response: Thank you very much for your valuable comments. Since the data did not follow a normal distribution, we changed the Pearson correlation analysis to a Spearman correlation analysis. And we have added this content in the STATISTICAL ANALYSIS section, which you can find on page 11 of the manuscript.

Question 2. Steps for Spearman Correlation Analysis.

Spearman correlation is a non-parametric method that evaluates monotonic relationships by ranking the data. Please indicate the exact steps undertaken to implement it in your analysis. For instance, note any rank transformations or adjustments for tied ranks, as these are required for this method.

Response: Thank you for your valuable comments. We have added detailed steps on Spearman correlation coefficient in the STATISTICAL ANALYSIS section on page 11 of the manuscript.

Question 3. Expanded Explanation for HACA Results.

Please elaborate further on the HACA (Hierarchical Cluster Analysis) results on page 24, sentence after Table 6. Specifically:

(a) Primary Clusters (broad groups): Discuss how similar sources (e.g., Cu and Hg from industrial activities) or comparable environmental behavior and mobilization processes (e.g., Pb, Ni, Zn, and Cr adsorbing onto organic matter or clay minerals) could explain these groupings.

(b) Sub-Clusters (Finer Groupings): Explain how these may reflect either geochemical similarities (e.g., Cr and Ni) or distinct combinations of anthropogenic and natural contributions (e.g., Cd and Pb arising from industrial emissions with varying inputs from transportation).

(c) Please revise the STATISTICAL ANALYSIS section to indicate that HACA was performed.

Response: Thank you very much for your comments. We have supplemented the information about HACA in the DISCUSSION section, described the primary clusters and sub-clusters, and discussed possible sources. In the STATISTICAL ANALYSIS section, we indicated that this study used HACA.

We have carefully reviewed the comments provided by the reviewers and made the necessary modifications and explanations. We have diligently worked to enhance the manuscript by implementing specific changes. These alterations do not affect the content or structure of the paper.

We sincerely appreciate the diligent efforts of Editors and Reviewers, and we hope that the revision will meet with approval.

Once again, thank you very much for your comments and suggestions.

Yours sincerely,

Liping Zhang

---

## [Editor Report · Decision Letter 3]

19 Jan 2025

Assessment of heavy metal pollution in sediments from the urban section of Yihe River, Linyi City, China

PONE-D-24-25973R3

Dear Dr. Zhang,

We’re pleased to inform you that your manuscript has been judged scientifically suitable for publication and will be formally accepted for publication once it meets all outstanding technical requirements.

Kind regards,

Timothy Omara

Academic Editor

PLOS ONE
---

## [Editor Report · Acceptance letter]

PONE-D-24-25973R3

PLOS ONE

Dear Dr. Zhang,

I'm pleased to inform you that your manuscript has been deemed suitable for publication in PLOS ONE. Congratulations! Your manuscript is now being handed over to our production team.

Kind regards,

on behalf of

Dr. Timothy Omara

Academic Editor

PLOS ONE